# PRIOR PREFERENCE LEARNING FROM EXPERTS: DESIGNING A REWARD WITH ACTIVE INFERENCE

## ABSTRACT

Active inference may be defined as Bayesian modeling of a brain with a biologically plausible model of the agent. Its primary idea relies on the free energy principle and the prior preference of the agent. An agent will choose an action that leads to its prior preference for a future observation. In this paper, we claim that active inference can be interpreted using reinforcement learning (RL) algorithms and find a theoretical connection between them. We extend the concept of expected free energy (EFE), which is a core quantity in active inference, and claim that EFE can be treated as a negative value function. Motivated by the concept of prior preference and a theoretical connection, we propose a simple but novel method for learning a prior preference from experts. This illustrates that the problem with inverse RL can be approached with a new perspective of active inference. Experimental results of prior preference learning show the possibility of active inference with EFE-based rewards and its application to an inverse RL problem.

## 1 INTRODUCTION

Active inference (Friston et al., 2009) is a theory emerging from cognitive science using a Bayesian modeling of the brain function (Friston et al., 2006; Friston, 2010; Friston et al., 2015; 2013), predictive coding (Friston et al., 2011; Lopez-Persem et al., 2016), and the free energy principle (Friston, 2012; Parr & Friston, 2019; Friston, 2019). It states that the agents choose actions to minimize an expected future surprise (Friston et al., 2012; 2017a;b), which is a measurement of the difference between an agent's prior preference and expected future. Minimization of an expected future surprise can be achieved by minimizing the *expected free energy* (EFE), which is a core quantity of active inference. Although active inference and EFE have been inspired and derived from cognitive science using a biologically plausible brain function model, its usage in RL tasks is still limited owing to its computational issues and prior-preference design. (Millidge, 2020; Fountas et al., 2020)

First, EFE requires heavy computational cost. A precise computation of an EFE theoretically averages all possible policies, which is clearly intractable as an action space $\mathcal{A}$ and a time horizon $T$ increase in size. Several attempts have been made to calculate the EFE in a tractable manner, such as limiting the future time horizon from $t$ to $t + H$ (Tschantz et al., 2019), and applying Monte-Carlo based sampling methods (Fountas et al., 2020; Çatal et al., 2020) for the search policies.

Second, it is unclear how the prior preferences should be set. This is the same question as how to design the rewards in the RL algorithm. In recent studies (Fountas et al., 2020; Çatal et al., 2020; Ueltzhöffer, 2018) the agent's prior preference is simply set as the final goal of a given environment for every time step. There are some environments in which the prior preference can be set as time independent. However, most prior preferences in RL problems are neither simple nor easy to design because prior preferences of short and long-sighted futures should generally be treated in different ways.

In this paper, we first claim that there is a theoretical connection between active inference and RL algorithms. We then propose prior preference learning (PPL), a simple and novel method for learning a prior-preference of an active inference from an expert simulation. In Section 2, we briefly introduce the concept of an active inference. From the previous definition of the EFE of a deterministic policy, in Section 3, we extend the previous concepts of active inference and theoretically demonstrate that it can be analyzed in view of the RL algorithm. We extend this quantity to a stochastic

policy network and define an action-conditioned EFE for a given action and a given policy network. Following Millidge (2020) using a bootstrapping argument, we show that the optimal distribution over the first-step action induced from active inference can be interpreted using Q-Learning. Consequently, we show that EFE can be treated as a negative value function from an RL perspective. From this connection, in Section 4, we propose a novel inverse RL algorithm for designing EFE-based rewards, by learning a prior preference from expert demonstrations. Through such expert demonstrations, an agent learns its prior preference given the observation to achieve a final goal, which can effectively handle the difference between local and global preferences. It will extend the scope of active inference to inverse RL problem. Our experiments in Section 6 show the applicability of active inference based rewards using EFE to an inverse RL problem.

## 2 ACTIVE INFERENCE

The active inference environment rests on the partially observed Markov decision process settings with an observation that comes from sensory input $o_t$ and a hidden state $s_t$ which is encoded in the agent's latent space. We will discuss a continuous observation/hidden state space, a discrete time step, and a discrete action space $\mathcal{A}$: At a current time $t < T$ with a given time horizon $T$, the agent receives an observation $o_t$. The agent encodes this observation to a hidden state $s_t$ in its internal generative model, (i.e., a generative model for the given environment in an agent) and then searches for the action sequence that minimizes the *expected future surprise* based on the agent's prior preference $\tilde{p}(o_\tau)$ of a future observation $o_\tau$ with $\tau > t$. (i.e. The agent avoids an action which leads to unexpected and undesired future observations, which makes the agent surprised.)

In detail, we can formally illustrate the active inference agent's process as follows: $s_t$ and $o_t$ are a hidden state and an observation at time $t$, respectively. In addition, $\pi = (a_1, a_2, ..., a_T)$ is a sequence of actions. Let $p(o_{1:T}, s_{1:T})$ be a generative model of the agent with its transition model $p(s_{t+1}|s_t, a_t)$, and $q(o_{1:T}, s_{1:T}, \pi)$ be a variational density. A distribution over policies $q(\pi)$ will be determined later. From here, we can simplify the parameterized densities as trainable neural networks with $p(o_t|s_t)$ as a decoder, $q(s_t|o_t)$ as an encoder, and $p(s_{t+1}|s_t, a_t)$ as a transition network in our generative model.

First, we minimize the current surprise of the agent, which is defined as $-\log p(o_t)$. Its upper bound can be interpreted as the well-known negative ELBO term, which is frequently referred to as the variational free energy $F_t$ at time $t$ in studies on active inference.

$$-\log p(o_t) \leq \mathbb{E}_{q(s_t|o_t)}[\log q(s_t|o_t) - \log p(o_t, s_t)] = F_t \tag{1}$$

Minimizing $F_t$ provides an upper bound on the current surprise, and makes our networks in the generative model well-fitted with our known observations of the environment and its encoded states. For the future action selection, the total EFE $\mathcal{G}(s_t)$ over all possible policies at the current state $s_t$ at time $t$ should be minimized.

$$\mathcal{G}(s_t) = \mathbb{E}_{q(s_{t+1:T}, o_{t+1:T}, \pi)}[\log \frac{q(s_{t+1:T}, \pi)}{\tilde{p}(s_{t+1:T}, o_{t+1:T})}] \tag{2}$$

Focusing on the distribution $q(\pi)$, it is known that the total EFE $\mathcal{G}(s_t)$ is minimized when the distribution over policies $q(\pi)$ follows $\sigma(-G_\pi(s_t))$, where $\sigma(\cdot)$ is a softmax over the policies and $G_\pi(s_t)$ is the EFE under a given state $s_t$ for a fixed sequence of actions $\pi$ at time $t$. (Millidge et al., 2020)

$$G_\pi(s_t) = \sum_{\tau > t} G_\pi(\tau, s_t) = \sum_{\tau > t} \mathbb{E}_{q(s_\tau, o_\tau|\pi)}[\log \frac{q(s_\tau|\pi)}{\tilde{p}(o_\tau)q(s_\tau|o_\tau)}] \tag{3}$$

This means that a lower EFE is obtained for a particular action sequence $\pi$; a lower future surprise will be expected and a desired behavior $\tilde{p}(o)$ will be obtained. Several active inference studies introduce a temperature parameter $\gamma > 0$ such that $q_\gamma(\pi) = \sigma(-\gamma G_\pi(s_t))$ to control the agent's behavior between exploration and exploitation. Because we know that the optimal distribution over the policies is $q(\pi) = \sigma(-G_\pi(s_t))$, the action selection problem boils down to a calculation of the expected free energy $G_\pi(s_t)$ of a given action sequence $\pi$.

The learning process of active inference contains two parts: (1) learning an agent's generative model with its trainable neural networks $p(o_t|s_t)$, $q(s_t|o_t)$, and $p(s_{t+1}|s_t, a_t)$ that explains the current observations and (2) learning to select an action that minimizes a future expected surprise of the agent by calculating the EFE of a given action sequence.

## 3 EFE AS A NEGATIVE VALUE: BETWEEN RL AND ACTIVE INFERENCE

In this section, we first extend the definition of an EFE to a stochastic policy. We then, propose an action-conditioned EFE that has a similar role as a negative action-value function in RL. Based on these extensions, we will repeat the arguments in the active inference and claim that the RL algorithm with EFE as a negative value is equivalent to controlling the policy network toward an ideal distribution in the active inference. Calculating the expected free energy $G_\pi(s_t)$ for all possible deterministic policies $\pi$ is intractable even in a toy-example task because the number of policies rapidly increases as the time horizon $T$ and the number of actions $|\mathcal{A}|$ increases. Instead of searching for a number of deterministic policies, we extend the concept of EFE to a stochastic policy based on a policy network $\phi = \phi(a_t|s_t)$. In this case, we also extend $q(s_\tau|\pi)$ to $q(s_\tau, a_{\tau-1}|\phi) := p(s_\tau|s_{\tau-1}, a_{\tau-1})\phi(a_{\tau-1}|s_{\tau-1})$, which is a distribution over the states and actions, where each action $a_{\tau-1}$ is only dependent on state $s_{\tau-1}$ with $\phi(a_{\tau-1}|s_{\tau-1})$. Plugging in a deterministic policy $\phi = \pi$ yields $q(s_\tau, a_{\tau-1}|\pi) = p(s_\tau|s_{\tau-1}, a')$ with $\pi(s_{\tau-1}) = a'$, which is the same equation used in previous studies on active inference. Suppose we choose an action $a_t$ based on the current state $s_t$ and a given action network $\phi$, its corresponding expected free energy term can then be written as follows. This can be interpreted as an EFE of a sequence of policies $(\phi, \phi, ..., \phi)$ by substituting our extensions for the probabilities in (3).

$$G_\phi(s_t) = \mathbb{E}_{\prod_{\tau>t} q(s_\tau, o_\tau, a_{\tau-1}|\phi)}[\sum_{\tau>t} \log \frac{q(s_\tau, a_{\tau-1}|\phi)}{\tilde{p}(o_\tau)q(s_\tau|o_\tau)}] \tag{4}$$

Note that the agent uses the same action network $\phi(a_\tau|s_\tau)$ for all $\tau$, and we can therefore rewrite this equation in a recursive form.

$$
\begin{aligned}
G_\phi(s_t) &= \mathbb{E}_{\prod_{\tau>t} q(s_\tau, o_\tau, a_{\tau-1}|\phi)}[\sum_{\tau>t} \log \frac{q(s_\tau, a_{\tau-1}|\phi)}{\tilde{p}(o_\tau)q(s_\tau|o_\tau)}] \\
&= \mathbb{E}_{\prod_{\tau>t} q(s_\tau, o_\tau, a_{\tau-1}|\phi)}[\log \frac{q(s_{t+1}, a_t|\phi)}{\tilde{p}(o_{t+1})q(s_{t+1}, a_t|o_{t+1})} + \sum_{\tau>t+1} \log \frac{q(s_\tau, a_{\tau-1}|\phi)}{\tilde{p}(o_\tau)q(s_\tau|o_\tau)}] \\
&= \mathbb{E}_{q(s_{t+1}, o_{t+1}, a_t|\phi)}[\log \frac{q(s_{t+1}, a_t|\phi)}{\tilde{p}(o_{t+1})q(s_{t+1}|o_{t+1})} + \mathbb{E}_{\prod_{\tau>t+1} q(s_\tau, o_\tau, a_{\tau-1}|\phi)} \sum_{\tau>t+1} \log \frac{q(s_\tau, a_{\tau-1}|\phi)}{\tilde{p}(o_\tau)q(s_\tau|o_\tau)}] \\
&= \mathbb{E}_{q(s_{t+1}, o_{t+1}, a_t|\phi)}[\log \frac{q(s_{t+1}, a_t|\phi)}{\tilde{p}(o_{t+1})q(s_{t+1}|o_{t+1})} + G_\phi(s_{t+1})]
\end{aligned}
\tag{5}
$$

Replacing and fixing the first action $a_t = a$, an action-conditioned EFE $G_\phi(s_t|a)$ of a given action $a$ and a policy network $\phi$ is then defined as follows:

$$
\begin{aligned}
G_\phi(s_t|a) &:= \mathbb{E}_{q(s_{t+1}, o_{t+1}, a_t|a_t=a)}\left[\log \frac{q(s_{t+1}, a_t|a_t=a)}{\tilde{p}(o_{t+1}, s_{t+1})} + G_\phi(s_{t+1})\right] \\
&= \mathbb{E}_{p(s_{t+1}|s_t, a_t=a)p(o_{t+1}|s_{t+1})}\left[\log \frac{p(s_{t+1}|s_t, a_t=a)}{\tilde{p}(o_{t+1})q(s_{t+1}|o_{t+1})} + G_\phi(s_{t+1})\right]
\end{aligned}
\tag{6}
$$

Taking an expectation over $\phi(a|s_t)$, we obtain the relationship between $G_\phi(s_t)$ and $G_\phi(s_t|a_t)$.

$$\mathbb{E}_{\phi(a|s_t)}[G_\phi(s_t|a)] = G_\phi(s_t) + \mathcal{H}[\phi(a|s_t)] \tag{7}$$

We may consider separating the distribution over the first-step action from $\phi$ to find an alternative distribution that minimizes the EFE. Substituting the distribution over the first-step action as $q(a_t)$ instead of $\phi(a_t)$, we obtain its one-step substituted EFE as indicated below. This can be interpreted as the EFE of a sequence of a policies $(q(a_t), \phi, ..., \phi)$

$$
\begin{aligned}
\mathcal{G}^\phi_{1-step}(s_t) &= \mathbb{E}_{q(a_t)q(s_{t+1}, o_{t+1}|a_t) \prod_{\tau>t+1} q(s_\tau, o_\tau|\phi)}[\log \frac{q(a_t)q(s_{t+1}|a_t)q(s_{t+2:T}, a_{t+1:T-1}|\phi)}{\tilde{p}(s_{t+1:T}, o_{t+1:T})}] \\
&= \mathbb{E}_{q(a_t)q(s_{t+1}, o_{t+1}|a_t) \prod_{\tau>t+1} q(s_\tau, o_\tau|\phi)}[\log q(a_t) + \log \frac{q(s_{t+1}|a_t)}{\tilde{p}(o_{t+1}, s_{t+1})} + \log \prod_{\tau>t+1} \frac{q(s_\tau, a_{\tau-1}|\phi)}{\tilde{p}(o_\tau)q(s_\tau|o_\tau)}] \\
&= \mathbb{E}_{q(a_t)}\left[\log q(a_t) + G_\phi(s_t|a_t)\right]
\end{aligned}
\tag{8}
$$

Under a given $\phi$, the value above depends only on the distribution $q(a_t)$. Thus, minimizing the quantity above will naturally introduce the distribution $q^*(a_t) = \sigma_a(-G_\phi(s_t|a_t))$, which is known to be similar in terms of active inference to the $\gamma = 1$ case.

$$\mathcal{G}^\phi_{1-step}(s_t) = \mathbb{E}_{q(a_t)}\left[\log q(a_t) + G_\phi(s_t|a_t)\right] = KL(q(a_t)||q^*(a_t)) - \log \sum_{a_t \in \mathcal{A}} \exp(-G_\phi(s_t|a_t))$$
(9)

Therefore, $\mathcal{G}^\phi_{1-step}(s_t) \geq -\log \sum_{a_t \in \mathcal{A}} \exp(-G_\phi(s_t|a_t))$, and the equality holds if and only if $q(a_t) = \sigma_a(-G_\phi(s_t|a_t))$. Let us further consider the temperature hyperparameter $\gamma > 0$ such that $q^*_\gamma(a_t) = \sigma_a(-\gamma G(s_t|a_t))$. Similarly to the above, we obtain the following:

$$\begin{aligned}
\mathcal{G}^\phi_{1-step}(s_t) &= \frac{1}{\gamma}\mathbb{E}_{q(a_t)}\left[\gamma \log q(a_t) + \gamma G_\phi(s_t|a_t)\right] \\
&= \frac{1}{\gamma}\mathbb{E}_{q(a_t)}\left[(1-\gamma)(-\log q(a_t)) + \log q(a_t) - \log q^*_\gamma(a_t)\right] - \frac{1}{\gamma}\log \sum_{a_t \in \mathcal{A}} \exp(-\gamma G_\phi(s_t|a_t)) \\
&= (\frac{1}{\gamma} - 1)\mathcal{H}(q(a_t)) + \frac{1}{\gamma}KL(q(a_t)||q^*_\gamma(a_t)) - \frac{1}{\gamma}\log \sum_{a_t \in \mathcal{A}} \exp(-\gamma G_\phi(s_t|a_t))
\end{aligned}$$
(10)

From the arguments above, we can conclude that (1) $\mathcal{G}^\phi_{1-step}(s_t)$ is minimized when $q^*(a_t) = \sigma_a(-G(s_t|a_t))$, a 'natural case' in which $\gamma = 1$, which was heuristically set in the experiments described in Millidge (2020). (2) Plugging $q^*_\gamma(a_t) = \sigma_a(-\gamma G(s_t|a_t))$ to $q(a)$ in the equation above, we obtain

$$\begin{aligned}
\mathcal{G}^\phi_{1-step}(s_t) &= (1-\gamma)\mathbb{E}_{q^*_\gamma(a)}[G_\phi(s_t|a_t)] - \log D_\gamma \\
&\approx (1-\gamma)\mathbb{E}_{q^*_\gamma(a)}[G_\phi(s_t|a_t)] + \gamma \min_{a_t \in \mathcal{A}}\{G_\phi(s_t|a_t)\}
\end{aligned}$$
(11)

where $D_\gamma = \sum_{a_t \in \mathcal{A}} \exp(-\gamma G_\phi(s_t|a_t))$, and the last comes from the smooth approximation to the maximum function of log-sum-exp. This approximation becomes accurate when the maximum is much larger than the others.

$$-\log D_\gamma = -\log \sum_{a_t \in \mathcal{A}} \exp(-\gamma G_\phi(s_t|a_t)) \approx -\max_{a_t \in \mathcal{A}}\{-\gamma G_\phi(s_t|a_t)\} = \gamma \min_{a_t \in \mathcal{A}} G_\phi(s_t|a_t) \quad (12)$$

When $\gamma = 1$, the quantity $\mathcal{G}^\phi_{1-step}(s_t)$ is minimized with $q^*_1(a_t) = \sigma_a(-G(s_t|a_t))$, and its minimum value can be approximated as $-\log D_1 \approx \min_{a_t \in \mathcal{A}}\{G_\phi(s_t|a_t)\}$. When $\gamma \to \infty$, $q^*_\gamma(a_t)$ can be considered as a deterministic policy seeking the smallest EFE, which leads to the following:

$$\mathcal{G}^\phi_{1-step}(s_t) = (\frac{1}{\gamma} - 1)\mathcal{H}(q^*_\gamma(a_t)) - \frac{1}{\gamma}\log \sum_{a_t \in \mathcal{A}} \exp(-\gamma G_\phi(s_t|a_t)) \approx \min_{a_t \in \mathcal{A}} G_\phi(s_t|a_t) \quad (13)$$

Note that Q-learning with a negative EFE can be interpreted as $q^*_\gamma$ with the case of $\gamma = \infty$. When $\gamma \searrow 0$, $q^*_\gamma(a_t)$ converges to a uniform distribution over the action space $\mathcal{A}$, and the temperature hyperparameter $\gamma$ motivates the agent to explore other actions with a greater EFE. Its weighted sum also converges to an average of the action-conditioned EFE.

Considering the optimal policy $\phi^*$ that seeks an action with a minimum EFE, we obtain the following:

$$\begin{aligned}
G_{\phi^*}(s_t) &= \min_a G_{\phi^*}(s_t|a) \\
&= \min_a \mathbb{E}_{p(s_{t+1}|s_t, a_t=a)p(o_{t+1}|s_{t+1})}\left[\log \frac{p(s_{t+1}|s_t, a_t=a)}{\tilde{p}(o_{t+1})q(s_{t+1}|o_{t+1})} + G_{\phi^*}(s_{t+1})\right]
\end{aligned}$$
(14)

This equation is extremely similar to the Bellman optimality equation. Here, $G_\phi(s_t|a)$ and $G_\phi(s_{t+1})$ correspond to the action-value function and the state value function, respectively. Based on this similarity, we can consider the first term $\log \frac{p(s_{t+1}|s_t, a_t=a)}{\tilde{p}(o_{t+1})q(s_{t+1}|o_{t+1})}$ as a one-step negative reward (because active inference aims to minimize the expected free energy of the future) and EFE as a negative value function.

## 4 PPL: PRIOR PREFERENCE LEARNING FROM EXPERTS

From the previous section, we verified that using $\log \frac{p(s_{t+1}|s_t,a_t)}{\tilde{p}(o_{t+1})q(s_{t+1}|o_{t+1})}$ as a negative reward can handle EFE with traditional RL methods. Through a simple calculation, the given term can be decomposed as follows:

$$R_t := -\log \frac{p(s_{t+1}|s_t,a_t)}{\tilde{p}(o_{t+1})q(s_{t+1}|o_{t+1})} = \log \tilde{p}(o_{t+1}) + (-\log \frac{p(s_{t+1}|s_t,a_t)}{q(s_{t+1}|o_{t+1})}) = R_{t,i} + R_{t,e} \quad (15)$$

The first term measures the similarity between a preferred future and a predicted future, which is an intuitive value. The second term is called the epistemic value, which encourages the exploration of an agent. (Friston et al., 2015) The epistemic value can be computed either with prior knowledge on the environment of an agent or with various algorithms to learn a generative model. (Igl et al., 2018; Kaiser et al., 2019)

The core key to calculating the one-step reward and learning EFE is the prior preference $\tilde{p}(o_t)$, which includes information about the agent's preferred observation and goal. Setting a prior preference for an agent is a challenging point for active inference. A simple method that has been used in recent studies (Ueltzhöffer, 2018; Çatal et al., 2020) is to set a prior preference as a Gaussian distribution with mean of the goal position. However, it would be inefficient to use the same prior preference for all observations $o$. We called this type of preference *global preference*. Taking a mountain-car environment as an example, to reach the goal position, the car must move away from the goal position in an early time step. Therefore $\tilde{p}(o_t)$ must contain information about *local preference*. It is also difficult to directly design the prior preference for some observation space, such as in an image.

To solve the problems arising from active inference, we introduce prior preference learning from experts (PPL). Suppose that an agent can access expert simulations $S = \{(o_{i,1}, ..., o_{i,T}\}_{i=1}^N$, where $N$ is the number of simulations. From the expert simulations $S$, an agent can learn an expert's prior $p(o_{t+1}|o_t)$ based on the current observation $o_t$. Model $p(o_{t+1}|o_t)$ captures the expert's local preference, which is more effective than the global preference. We can state that this method can be applied only with expert simulations, regardless of the knowledge on the forward dynamics of the environment.

Once learning the prior preference $p(o_{t+1}|o_t)$, we can calculate $R_t$ given in (15). We can use any RL alogorithm to approximate EFE, $G_{\phi^*}(s_t)$ in (14). In Algorithm 1, we provide our pseudo-code of PPL with Q-learning.

## 5 RELATED WORKS

**Active inference on RL.** Our works are based on active inference and reinforcement learning. Active inference was first introduced in Friston et al. (2006), inspired from neuroscience and free energy principle. It explains how a biological system is maintained with a brain model. Furthermore, Friston et al. (2009) treated the relation between active inference and reinforcement learning. Early studies (Friston, 2010; Friston et al., 2011; 2015; 2017a) dealt with a tabular or model based problem due to computational cost.

Recently, Ueltzhöffer (2018) introduced deep active inference which utilizes deep neural network to approximate the observation and transition models on MountainCar environment. This work used EFE as an objective function and its gradient propagates through the environment dynamics. To handle this problem, Ueltzhöffer (2018) used stochastic weights to apply an evolutionary strategy (Salimans et al., 2017), which is a method for gradient approximation. For a stable gradient approximation, about $10^4$ environments were used in parallel.

Çatal et al. (2020); Millidge (2020) introduced end-to-end differentiable models by including environment transition models. Both require much less interaction with environments than before. Çatal et al. (2020) used the Monte Carlo sampling to approximate EFE and used global prior preference. On the other hand Millidge (2020) used bootstrapping methods and used common RL rewards with model driven values derived from EFE. Millidge (2020) verified that the model driven values induce faster and better results on MountainCar environment. Furthermore, Tschantz et al. (2020); Millidge et al. (2020) introduced a new KL objective called free energy of expected future (FEEF) which is related to probabilistic RL (Levine, 2018b; Rawlik, 2013; Kappen et al., 2012; Lee et al., 2019).

---

**Algorithm 1:** Inverse Q-Learning with Prior Preference Learning

---

Learning prior preference from expert simulations;

**Input:** Expert simultaions $S = (o_{i,1}, ...o_{i,T})_{i=1}^{N}$

Initialize a prior preference network $\tilde{p}_\theta(o)$;

**while** *not converge* **do**

   | Compute loss $L_{ppl}(\tilde{p}(o_t), o_{t+1})$;
   | Update $\theta \leftarrow \theta - \alpha \nabla L_{ppl}$

**end**

**Output:** $\tilde{p}_\theta(o)$

Learning EFE and forward dynamic of an environment;

**Input:** Prior preference $\tilde{p}_\theta(o)$

Initialize the forward dynamic $p_\eta(o|s), T_\eta(s_{t+1}|s_t, a_t), q_\eta(s|o)$, and EFE network $G_\xi(s_t, a_t)$.

  **while** *not converge* **do**

    Reset environment;

    **for** $t \leftarrow 0$ **do**

       Select action $a_t = \arg\max_a G(s_t, a)$ with $\epsilon$-greedy;

       Observe new observation $o_{t+1}$;

       Compute $R_t = \log \frac{T(s_{t+1}|s_t, a_t)}{\tilde{p}(o_{t+1})q(s_{t+1}|o_{t+1})}$;

       Compute the environment model loss $L_{model}((p \circ T \circ q)(o_t), o_{t+1})$;

       Compute the EFE network loss $L_{efe}(G_\xi(s_t, a_t), R_t + \max_a G_\xi(s_{t+1}, a))$;

       Update $\eta \leftarrow \eta - \alpha \nabla L_d$ and $\xi \leftarrow \xi - \alpha \nabla L_e fe$.

    **end**

**end**

---

Our work extends the scope of active inference to inverse RL by learning a preferred observation from experts. A common limitation of previous studies on active inference is the ambiguity of prior preference distribution. Previous works have done in environments where its prior preference can be naturally expressed, which is clearly not true in general. PPL is an active inference based approach for the invser RL problem setting, which allows us to learn a prior preference from expert simulations. This broadens the scope of active inference and provides a new perspective about a connection between active inference and RL.

**Control as inference.** Levine (2018a) proposed *control as inference* framework, which interprets a control problem as a probabilistic inference with an additional binary variable $\mathcal{O}_t$ that indicates whether given action is optimal or not. Several studies on the formulation of RL as an inference problem (Todorov, 2008; Kappen et al., 2009) have been proposed. Control as inference measures a probability that a given action is optimal based on a given reward with $p(\mathcal{O}_t = 1|s_t, a_t) = \exp(r(s_t, a_t))$. That is, from the given reward and the chosen probability model, control as inference calculates the probability of the given action $a_t$ with the state $s_t$ to be optimal. In contrast, active inference measures this probability as in (15) with a prior preference distribution $\tilde{p}$ and constructs a reward, viewing EFE as a negative value function. Although control as inference and active inference as a RL in Section 3 have a theoretical similarity based on a duality of a control problem and an inference, our proposed PPL can interpret the reward $r(s_t, a_t)$ and the message function $\beta(s_t)$ in Levine (2018a) as EFE-related quantities. Therefore, learning the expert as a prior preference $\tilde{p}$ immediately constructs its active inference based reward function and thereby applicable to a reward construction problem and several inverse RL problems.

## 6 EXPERIMENTS

In this section, we discuss and compare the experimental results of our proposed algorithm PPL and other inverse RL algorithms on the several classical control environments. We evaluate our approach with a classic control environment implemented in Open AI Gym (Brockman et al., 2016). First, we aim to compare the conventional global preference method to the PPL. We expect the PPL to be effective in environments where the local and global preferences are different. Second, we claim

Table 1: Experment setting for PPL and global preference and its variants

|  | Preference | Batch sampling | Reward |
|---|---|---|---|
| Setting 1 | PPL | Replay memory + Experts | $R_{t,i} + R_{t,e}$ |
| Setting 2 | PPL | Replay memory | $R_{t,i} + R_{t,e}$ |
| Setting 3 | PPL | Replay memory + Experts | $R_{t,i}$ |
| Setting 4 | Global preference | Replay memory + Experts | $R_{t,i} + R_{t,e}$ |

that our active inference based approach can achieve a compatible results with current inverse RL algorithms. Expert simulations were obtained from Open AI RL baseline zoo (Raffin, 2018).

## 6.1 PPL AND GLOBAL PREFERENCE

First, we compare our proposed algorithm (setting 1 in Table 1) and its variants. Table 1 contains four experimental settings, where the setting 1 is our proposed method and the others are experimental groups to compare the effects of an expert batch, the epistemic value $R_{t,e}$, and PPL.

**Expert Batch.** We use expert simulations for batch sampling when learning EFE and a forward dynamic model. This allows an EFE network and a dynamic model to be trained even in states that do not reach the early stage of learning. We use the Q-learning algorithm to learn the EFE network. Commonly used techniques in RL, such as replay memory and target network (Mnih et al., 2015) are used.

**Reward.** We observe how an epistemic value $R_{t,e}$ in (15) influences the learning process and its performance.

**Global preference.** Global preference is a distribution over a state which can be naturally induced from the agent's goal, whereas our PPL is a learned prior preference from the expert's simulations. Roughly, global preference can be understood as a hard-coded prior preference based on the prior knowledge of the environment, as a 'goal directed behavior' in Ueltzhöffer (2018). Detailed hard-coded global preferences in our experiments can be found in the below sub-subsection.

### 6.1.1 ENVIRONMENTS AND EXPERIMENT DETAILS

We tested three classical control environments: Acrobot, Cartpole, and MountainCar. We used 5000 pairs of $(o_t, o_{t+1})$ to train the prior preference $\tilde{p}(o_{t+1})$. We used the same neural network architecture for all environments, except for the number of input and output dimensions. During the training process, we clip the epistemic value to prevent a gradient explosion while using the epistemic value in settings 1, 2, and 4.

**Acrobot** is an environment with a two-link pendulum, where the second link is actuated. The goal of this environment is to swing the end effector up to the horizontal line. The state space has six dimensions: $(\cos\theta_1, \sin\theta_1, \cos\theta_2, \sin\theta_2, \dot{\theta}_1, \dot{\theta}_2)$, where $\theta_1$ and $\theta_2$ are the two rotational joint angles. The action space consists of three actions: accelerating +1, 0, or -1 torque on the joint between the two links. We did not run setting 4 in this study, because Acrobot is ambiguous in defining the global preference of the environment.

**Cartpole** is an environment with a cart and a pole on the cart. The goal is to balance the pole by controlling the cart. The state space is of 4 dimensions: $(x, \dot{x}, \theta, \dot{\theta})$, where $x$ is the position of the cart and $\theta$ is the angle of the pole from the vertical line. The action space consists of two actions: accelerating the cart to the left or right. The global preference is given by a Gaussian distribution with mean $(0, 0, 0, 0)$ with a fixed standard deviation $0.1$ on each dimension.

**MountainCar** is an environment with a car on the valley. The goal is to make the car reach the goal position, the right peak of the mountain. The state space is of two dimensions $(x, \dot{x})$ where $x$ is the position of the car. The action space consists of three actions: accelerating the car to the left or right, or leaving it. The global preference is given by a Gaussian distribution with mean $(0.5, 0)$ with a fixed standard deviation $0.1$ on each dimension. Note that $(0.5, 0)$ is the state of the goal position with zero velocity.

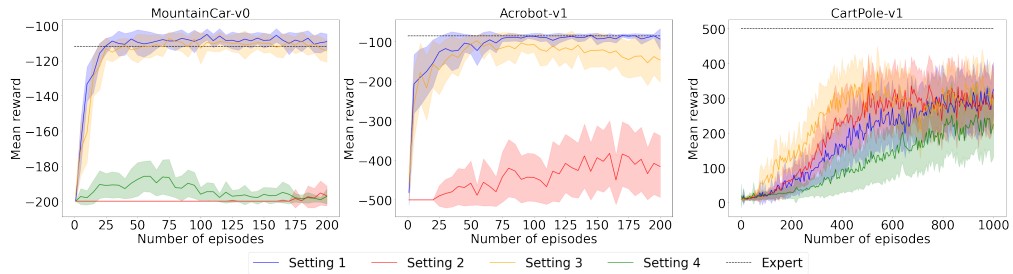

Figure 1: Experiment results on three classical control environments: MountainCar-v0, Acrobot-v1, and CartPole-v1. The curves in the figure were averaged over 50 runs, and the standard deviation of 50 runs is given as a shaded area. Each policy was averaged out of 5 trials. All rewards of the environment follow the default settings of Open AI Gym.

### 6.1.2 RESULTS AND DISCUSSIONS

Figure 1 shows the experimental results on MountainCar, Acrobot, and Cartpole. Their performances are compared and benchmarked with the default reward of the environments.

**PPL and global preference.** Comparing setting 1 (blue line) and setting 2 (red line), it can be seen that PPL is more efficient than the conventional global preference as expected. In particular, in MountainCar, we can see that little learning is achieved. This seems to be because the difference between global and expert preferences is greater in the MountainCar environment. In the Cartpole, setting 2 learned more slowly than setting 1.

**Expert Batch.** Comparing setting 1 (blue line) and setting 3 (orange line), it can be seen that using the expert batch is helpful for certain tasks. With Acrobot and MountainCar, the use of an expert batch performs better than the case without an expert batch. However, the results without expert batch are marginally better than those of setting 1 for Cartpole. This is because an agent of Cartpole only moves near the initial position, and thus there is no need for an expert batch to discover the dynamics of the generative model.

**Epistemic Value.** We found that the epistemic value in the EFE term does not significantly impact the training process. Comparing setting 1 (blue line) and setting 4 (green line), the results were similar regardless of whether the epistemic value was used. In Acrobot and MountainCar, standard deviations were marginally smaller, but there were no significant differences between them. In the result of CartPole-v1, we found that setting 4 with no epistemic value term learned faster than our proposed setting 1 at the beginning of the training process. We deduce that this initial performance drop is due to the instability of the epistemic term. At the beginning of the training process, the generative model is not learned, and thus the related epistemic term becomes unstable. We leave this issue to a future study.

### 6.2 PPL AND INVERSE RL ALGORITHMS

Second, we check that our proposed PPL is compatible with traditional inverse RL algorithms. We compared PPL with behavioral cloning (BC, Pomerleau (1989)) and maximum entropy inverse RL (MaxEnt, Ziebart et al. (2008)) as benchmark models. We use setting 1 in Table 1 as our proposed PPL here, and we test on MountainCar-v0 and CartPole-v1. Note that BC does not need to interact with the environment and the state space was discretized to use original MaxEnt algorithm. We also tried to run the experiment on Acrobot-v1 for PPL and other benchmarks, but we failed to make the agent learn with MaxEnt. A discretized state space for MaxEnt becomes larger exponentially to its state dimension. We think it is due to a larger dimension of its state space compared to the others. Therefore, we only report that PPL and BC give similar results to Acrobot-v1.

We verified that our method PPL gives compatible results on the MountainCar-v0 and CartPole-v1. Compared to MaxEnt, PPL shows better results than MaxEnt on both environments. Note that MaxEnt needs much more episodes to converge. Also, PPL obtained almost similar mean rewards to BC on MountainCar-v0, whereas BC gives better results than PPL on CartPole-v1.

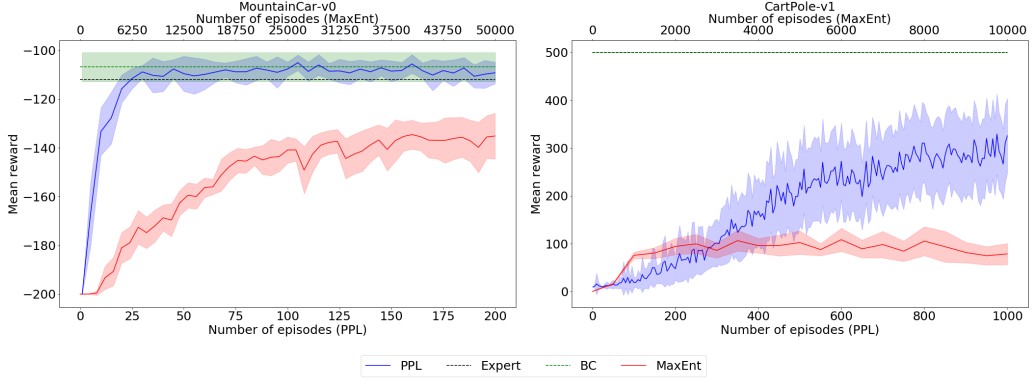

Figure 2: Inverse RL Experiment results on MountainCar-v0 (left) and CartPole-v1 (right). The curves in the figure were averaged over 50 runs, and the standard deviation of 50 runs is given as a shaded area. Note that black and green dashed line on the right are overlapped. All rewards of the environment follow the default settings of Open AI Gym. (BC : Behavioral Cloning, MaxEnt : Maximum Entropy)

## 7 CONCLUSION

In this paper, we introduced the use of active inference from the perspective of RL. Although active inference emerged from the Bayesian model of cognitive process, we show that the concepts of active inference, especially for EFE, are highly related to RL using the bootstrapping method. The only difference is that, the value function of RL is based on a reward, while active inference is based on the prior preference. We also show that active inference can provide insights to solve the inverse RL problems. Using expert simulations, an agent can learn a local prior preference, which is more effective than the global preference. Furthermore, our proposed active inference based reward with a prior preference and a generative model makes the previous invser RL problems free from an ill-posed state. Our work on active inference is complementary to RL because it can be applied to model-based RL for the design of reward and model-free RL for learning of generative models.

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
