# OpenReview forum: "Prior Preference Learning From Experts: Designing A Reward with Active Inference"
_ICLR.cc/2021/Conference — Reject_

### Official Review · AnonReviewer1 · 2020-10-28
**Interesting start but the message got lost.**

**Rating:** 5
**Confidence:** 4

**Review:**

The work in this paper draws connections between the active inference literature and reinforcement learning frameworks. The paper proposes a connection between these two methods more formally so that you can convert the active inference learning problem into a reinforcement learning problem. The paper also shows some success in being able to solve some simple control problems by providing some expert demonstrations it seems. It's not clear if this work is significantly different than the deep active inference paper which does something very similar and also runs experiments on the mountain car problem.

https://arxiv.org/abs/1709.02341

- The equations are not numbered in the paper, but the first equation in section 3 is a little unclear given the paragraph before it on how it would be obvious that this follows. It will help the reader to add more description about this.
- Q appears to be overloaded many times in the mathematics of the paper and makes it a bit difficult to follow the theory and section 2.
- The author's note and the experiment section that some type of expert preferences is greater than some type of global preference. This terminology is confusing and it's not clear where the "global preference" comes from.
- Similarly, "Expert Batch" is used to help learning but there does not appear to be a definition for the expert batch. Is it related to the 5000 tuples collected early? Where does this expert data come from?

---- Post Discussion ----
The discussion with the authors improved my understanding of how the paper fits with recent work.

---

> ### Author Response · Authors · 2020-11-18
> **Response for AnonReviewer #1.**
>
> We sincerely you for your kind and thoughtful comments on our paper. The comments are really helpful to improve our work.
>
> Before proceeding our rebuttal, we would like to make it clear that our work is clearly different from [1], which was mentioned in the review.
>
> - The dynamic model of an environment in [1] is set to be unknown, which makes generative models cannot be trained in ordinary backpropagation. In order to overcome this gradient issue, [1] used an evolution algorithm (in [1], Section 3.4) to stochastically approximate the gradients with the normal prior.
>
> - In the mountain car experiment in [1], the goal position was set and hard-coded near the terminal time step as a prior preference. It is similar to the way we mention global prior preference in our work. Detailed global preferences can be found in Section 6.1.1, which is hard-coded and problem-specific.
>
> - On the other hand, in our work, the dynamic model of an environment is directly trained with a backpropagation process. We showed that our method (Setting 1 in Table 1) is more effective than global preference. (Setting 4 in Table 1) Also, our method can be used in general environments, especially where the global preference is not direct to design such as Acrobot.
>
> - Mainly, before learning the agent’s policy, our algorithm learns the prior preference from the expert simulation which depends on current observation, whereas the previous work [1] learns this prior preference from known hard-coded prior knowledge with an ad-hoc setting for a mountain-car environment.
>
> We agree that the capitalized Q is widely used as an action-value function in the RL literature. Using a small q as a variational density function would be helpful to make it clear for the reader who is familiar with RL context. We sincerely thank you for the kind suggestion on the notation and the typos.
>
> We added related works in a new section. (Section 5) Please refer the common response above for the revised contents of the paper.
>
> [1] Kai Ueltzhoffer. Deep active inference. ¨ Biol. Cybern., 112(6):547–573, December 2018. ISSN 0340-1200. doi: 10.1007/s00422-018-0785-7. URL https://doi.org/10.1007/ s00422-018-0785-7.

---

> > ### Comment · AnonReviewer1 · 2020-11-20
> > **Responce and related work appreciated**
> >
> > Thank you for these notes and detailed comparison to prior work, this was a weaker point in the paper that made it difficult to appreciate its novelty. The paper is much clearer now. Given the related work now that tends to work on more complex problems than the environments used in the paper it would be helpful to compare to more difficult problems from the OpenAIGym benchmarks to better understand the scalability of the method.
> >
> > It would also be good to comment on how the work in this paper is different from this paper that also connects RL to Active inference.
> > Tschantz, A., Millidge, B., Seth, A. K., & Buckley, C. L. (2020). Reinforcement Learning through Active Inference. arXiv preprint arXiv:2002.12636

---

> > > ### Author Response · Authors · 2020-11-21
> > > **About the issues on the experiment and the difference between our and the suggested paper**
> > >
> > > We remark that our model for the experiments is based on the POMDP model as with the theoretical setting of active inference, whereas most of previous works [2, 3, 4], are based on the MDP setting in their experiments. (These can be found in Section 4 of [2], Section 4 of [3], and Section 3.5 of [4]) Therefore, our experiments based on POMDP setting with encoder and decoder are more general, and it is not directly compatible with the previous studies. This can be checked in our attached code in the supplementary materials.
> > >
> > > Recent papers on active inference have run their experiments in a similar scale of ours. Most of benchmark environments remain to be classic control problems. Also, challenging tasks in some previous studies are still few, compared with current RL literatures. For instance, an active inference problem with a more complicated environment with large-scale image observations is not studied enough. We think that this point would be a common obstacle in the area of active inference, and this scalability issue will be an important future work for active inference and its applications to RL.
> > >
> > > ---
> > >
> > > Mainly, [2] proposed a new quantity called ‘Free Energy of an Expected Future’ (FEEF). In the perspective of ‘the connection between RL and Active Inference’, [2] claimed that the FEEF is an upper bound of a KL divergence between the generative model and the prior preference. By minimizing FEEF, the divergence will be also minimized. From this boundedness, [2] claimed that they suggested the homology between RL and Active Inference, based on the previous studies on probabilistic RL and Control as Inference.
> > >
> > > In contrast, our work used a well-known EFE. We constructed a reward-like quantity from one-step EFE and showed the similarities between RL with negative EFE and active inference. Quantities of active inference can be interpreted as a traditional value function and reward with the well-known Bellman optimality equation.
> > >
> > > ---
> > >
> > > Other detailed differences between [2] and ours are following.
> > > - We extend the concept of EFE to a stochastic policy and directly optimize the policy network, whereas [2] searches the best policy among candidate polices.
> > > - [2] suggests a new quantity FEEF and connects this quantity in perspective of RL with a global prior preference (in Appendix E of [2], a hard-coded, model-specific prior preference). Ours uses EFE and an expert simulation, then designs an active inference based reward (reward design) with our proposed algorithm PPL and an expert simulation.
> > >
> > > ---
> > >
> > > [2] Tschantz, A., Millidge, B., Seth, A. K., & Buckley, C. L. (2020). Reinforcement Learning through Active Inference. arXiv preprint arXiv:2002.12636
> > >
> > > [3] Beren Millidge. Deep active inference as variational policy gradients. Journal of Mathematical Psychology, 96:102348, 2020. ISSN 0022-2496. doi: https://doi.org/10.1016/j.jmp.2020.102348. URL http://www.sciencedirect.com/science/article/pii/S0022249620300298.
> > >
> > > [4] Tschantz, A., Baltieri, M., Seth, A. K., & Buckley, C. L. (2020, July). Scaling active inference. In 2020 International Joint Conference on Neural Networks (IJCNN) (pp. 1-8). IEEE.

---

> > > > ### Comment · AnonReviewer1 · 2020-11-25
> > > > **Reponse**
> > > >
> > > > These additional clarifications are appreciated on the provided related work.
> > > >
> > > > The recent paper "SMiRL: Surprise Minimizing Reinforcement Learning in Dynamic Environments" (https://arxiv.org/abs/1912.05510) uses a similar reward function in difficult environments. It also includes experiments on imitation learning. It would good to include this paper in your discussion as well. The methods appear similar.
> > > >
> > > > Still, I have upgraded my score, thanks to the author's discussion and improved connections to recent work.

---

### Official Review · AnonReviewer2 · 2020-10-29
**The EFE-based approach is interesting, but validation is not sufficient.**

**Rating:** 5
**Confidence:** 3

**Review:**

This paper provides a theoretical connection between active inference and reinforcement learning and develops a method that can find a prior preference from experts. The new theory is derived from the concept of expected free energy (EFE) based on the free-energy principle.  Simulation experiments were conducted, and the effect of the prior preference learning was demonstrated.

The theoretical contribution of the paper is to find the relationship between EFE and negative value function and proposed a prior preference learning method. The theoretical connection is insightful and interesting.

However, the originality of the proposed method itself is not clear from the theoretical and practical viewpoints.
In the experiment, they compared their method with a baseline method, i.e., global preference.
There is no comparison between the pre-existing baseline method.
Though the EFE-based approach is very interesting, the authors did not succeed in providing evidence of the advantage of the proposed method.
It is questionable if this experiment is suitable for evaluating the main argument of this paper.

Also, from the viewpoint of the information-theoretic approach to RL and the relation to the free energy principle, studies related to "control as inference" is worth mentioning.

- Levine, Sergey. "Reinforcement learning and control as probabilistic inference: Tutorial and review." arXiv preprint arXiv:1805.00909 (2018).
- Okada, Masashi, and Tadahiro Taniguchi. "Variational inference mpc for bayesian model-based reinforcement learning." Conference on Robot Learning. 2020.
- Hafner, Danijar, et al. "Action and perception as divergence minimization." arXiv preprint arXiv:2009.01791 (2020).

<Minor comments>

Capitalized Q is used for representing a variational density function. Q is often used in action-value function in the context of RL. If this is not equivalent to Q-function, it cannot be very clear. I think using q is a better choice.

In 5.1.1, "We did not run setting 2 in this study, because Acrobat is ambiguous in defining the global
preference of the environment."
-> This may be "setting 4."


The definition of "global preference" is not given. To my understanding, the term is not so well-known in the community of imitation and reinforcement learning. That should be defined. Because of this, what the experiment showed is unclear to potential readers.


In conclusion, they describe, "We also show that active inference can provide insights to solve the inverse RL problems." However, they did not provide any explicit discussion over "inverse RL." This is actually the second time they mention "inverse RL." The first one is just at the end of the introduction.
This should be explicitly mentioned if the authors put this statement in conclusion.

---

> ### Author Response · Authors · 2020-11-18
> **Response for Reviewer #2.**
>
> We sincerely thank you for your kind and thoughtful comments on our paper. The comments are indeed helpful to improve our work.
>
> It is worth mentioning the relationship between active inference based approach (including our work) and the concept of ‘control as inference’. We briefly added and reflected these related studies that you mentioned in Section 5.
> In order to strengthen the connection between our work and inverse reinforcement learning, we also added a relationship between our proposed algorithm PPL and the inverse RL in Section 4.
>
> As the reviewers kindly remind that the comparison between classic IRL algorithms is heavily necessary, we added an additional in Experiment part (Section 6.2) that compares our approach with behavioral cloning (BC) and maximum entropy inverse RL (MaxEnt).
>
> For the reviewer’s concerns on the details of the experiment and the lack of comparison between IRL algorithms and ours, we added a note on the revised contents in the experiment section (Section 6). Please refer the common response above for the note on the experiment.
>
> We agree that the capitalized Q is widely used as an action-value function in the RL literature. Using a small q as a variational density function would be helpful to make it clear for the reader who is familiar with RL context. We sincerely thank you for the kind suggestion on the notation and the typos.

---

### Official Review · AnonReviewer3 · 2020-10-30
**This paper provides a interesting theoretical connection between active inference and reinforcement learning. Impact could be made stronger by extending the experiments.**

**Rating:** 6
**Confidence:** 2

**Review:**

This paper provides a theoretical connection between active inference and reinforcement learning. The authors show that the concept of expected free energy (EFE) can be extended to a stochastic setting and propose an action-conditioned EFE that can be interpreted as the well-known RL Q-function. They also propose a prior preference learning approach to learn from expert demonstrations.

The paper sheds light on a novel interpretation of active inference from the point of view of RL and demonstrates a theoretical connection between the two.

However, the concepts of active inference should be more clearly introduced and some intuition should be provided. It is quite hard for a reader not truly familiar with the field to follow.
Also, the experiment section is lacking comparison with traditional RL algorithms.

A few comments:

In the derivations in page 4, some approximations are used. It would help to explain why these can be made.

In the experiment section, it is not clear which algorithms are compared. We can assume that PPL refers to Algorithm 1, although the latter is never referred to in the text. Also, it is not clear what "conventional global preference" refers to. Also,  it would help put things in perspective to compare the authors' approach to classic RL/IRL algorithms.

Minor typo: in page 2, section 2, paragraph 1, line 5: space missing before “The agent”

---

> ### Author Response · Authors · 2020-11-18
> **Response for AnonReviewer #3.**
>
> We sincerely thank you for your kind and thoughtful comments on our paper. The comments are really helpful to improve our work.
>
> For the comment on the basic concept of active inference and its intuition, we mainly developed the concept of active inference as a mathematical formulation rather than its biological intuitions and motivations. We also agree that these intuitions and motivations are necessary for the readability of the paper for those not familiar with the concept of active inference. We gave an intuitive explanation on the minimization of the future surprise in Section 2. Also, this can be found in the first paragraph of the introduction. (Section 1)
>
> For the reviewer’s concerns on the details of the experiment and the lack of comparison between IRL algorithms and ours, we added a note on the revised contents in the experiment section (Section 6). Please refer the common response above for the note on the experiment.

---

### Author Response · Authors · 2020-11-18
**Common Response - Notice on the revision #1 and the major changes**

We sincerely thank all reviewers’ kind and thoughtful comments to our paper. All the comments were fruitful and crucial to improve our initial submission. Mainly, there are two CORE changes in the revision #1:

(1) The additional section (Section 5) that introduces related works has been added.

The main concerns on the difference between our work and previous studies (control as inference, Deep Active Inference by Kai Ueltzhoffer, and several neural network approaches on active inference) have been raised, thus we introduce such related works and compare the differences between them in the new section.

(2) The comparison between Maximum Entropy IRL, Behavior Cloning, and our algorithm PPL has been added. Also, the detailed notes on the ‘Expert batch’ and ‘Global Preference’ have been added in Section 6. (Experiments)

We added a detailed comment on the explanation on the settings of the experiment: ‘Global Preference’ and ‘Expert Batch’ in Table 1. For the additional KL term which was excluded in Setting 3, it is required to check whether the additional KL term really acts as a regularize.

- Global Preference is a distribution over a state which can be naturally induced from the agent’s goal, whereas our PPL is a learned prior preference from the expert’s simulations. Detailed global preferences can be found in Section 6.1.1. We also denote that the global preference in our paper is previously suggested by [1] (in Section 3.3.5) as a ‘Goal Directed Behavior’, which explicitly hard-codes the inference dynamics. The definitions of these details were added and revised in the modified pdf file.

- A replay memory was used in our baseline models, which is a general technique for Q-learning. Without using an expert batch, we sample a batch only from the replay memory, which was indicated as ‘Replay memory’ in Table 1. During the learning process, our models only learned near the initial state. In order to overcome this limitation without a reward, we sample a half of batch from expert simulation and the other half from the replay memory. We called this technique as ‘Expert Batch’ in Table 1. We found that the details were in the second paragraph of Section 5 with less attention, which was revised in the modified pdf file.

As the reviewers kindly remind that the comparison between classic IRL algorithms is heavily necessary, we added an additional in Experiment part (Section 6.2) that compares our approach with behavioral cloning (BC) and maximum entropy inverse RL (MaxEnt).

Also, there are some minor changes in the revision #1:

- Capital Q in the derivation has been changed into small q, in order to avoid the confusion with Q-function in the RL literature.

- We added a numbering on equations.

- We’ve revised the minor typos that were pointed out by reviewers.

- We added a brief intuitive explanation on the active inference in Section 2.

We sincerely thank you all for your great efforts on the review of our paper.

[1] Kai Ueltzhoffer. Deep active inference. ¨ Biol. Cybern., 112(6):547–573, December 2018. ISSN 0340-1200. doi: 10.1007/s00422-018-0785-7. URL https://doi.org/10.1007/ s00422-018-0785-7.

---

### Decision · Program_Chairs · 2021-01-07
**Final Decision**

**Decision:**

Reject

**Comment:**

The meta-reviewer agrees with the reviewers that this is a marginal case. Conditioned on the quality of content and comparisons to other works:
Constrained Reinforcement Learning With Learned Constraints (https://openreview.net/forum?id=akgiLNAkC7P)
Parrot: Data-Driven Behavioral Priors for Reinforcement Learning (https://openreview.net/forum?id=Ysuv-WOFeKR)
PERIL: Probabilistic Embeddings for hybrid Meta-Reinforcement and Imitation Learning (https://openreview.net/forum?id=BIIwfP55pp)

We believe that the paper is not ready for publication yet. We would strongly encourage the authors to use the reviewers' feedback to improve the paper and resubmit to one of the upcoming conferences.